

# Identifying contamination with advanced visualization and analysis practices: metagenomic approaches for eukaryotic genome assemblies

Tom O. Delmont[1] and A. Murat Eren[1,2]

[1] Department of Medicine, University of Chicago, Chicago, IL, United States
[2] Josephine Bay Paul Center, Marine Biological Laboratory, Woods Hole, MA, United States

## ABSTRACT

High-throughput sequencing provides a fast and cost-effective mean to recover genomes of organisms from all domains of life. However, adequate curation of the assembly results against potential contamination of non-target organisms requires advanced bioinformatics approaches and practices. Here, we re-analyzed the sequencing data generated for the tardigrade *Hypsibius dujardini,* and created a holistic display of the eukaryotic genome assembly using DNA data originating from two groups and eleven sequencing libraries. By using bacterial single-copy genes, k-mer frequencies, and coverage values of scaffolds we could identify and characterize multiple near-complete bacterial genomes from the raw assembly, and curate a 182 Mbp draft genome for *H. dujardini* supported by RNA-Seq data. Our results indicate that most contaminant scaffolds were assembled from Moleculo long-read libraries, and most of these contaminants have differed between library preparations. Our re-analysis shows that visualization and curation of eukaryotic genome assemblies can benefit from tools designed to address the needs of today's microbiologists, who are constantly challenged by the difficulties associated with the identification of distinct microbial genomes in complex environmental metagenomes.

## INTRODUCTION

Advances in high-throughput sequencing technologies are revolutionizing the field of genomics by allowing researchers to generate large amount of data in a short period of time (*Loman & Pallen, 2015*). These technologies, combined with advances in computational approaches, help us understand the diversity and functioning of life at different scales by facilitating the rapid recovery of bacterial, archaeal, and eukaryotic genomes (*Venter et al., 2001*; *Schleper, Jurgens & Jonuscheit, 2005*; *Brown et al., 2015*). Yet, the recovery of genomes is not straightforward, and reconstructing bacterial and archaeal versus eukaryotic genomes present researchers with distinct pitfalls and challenges that result in different molecular and computational workflows.

For instance, difficulties associated with the cultivation of bacterial and archaeal organisms (*Schloss & Handelsman, 2003*) have persuaded microbiologists to reconstruct

Corresponding author
A. Murat Eren,
a.murat.eren@gmail.com,
meren@uchicago.edu

genomes directly from the environment through assembly-based metagenomics workflows and genome binning. This workflow commonly entails (1) whole sequencing of environmental genetic material, (2) assembly of short reads into contiguous DNA segments (contigs), and (3) identification of draft genomes by binning contigs that originate from the same organism. Due to the extensive diversity of bacteria and archaea in most environmental samples (*Gans, Wolinsky & Dunbar, 2005*; *Rusch et al., 2007*), the field of metagenomics has rapidly evolved to accurately delineate genomes in assembly results. Today, microbiologists often exploit two essential properties of bacterial and archaeal genomes to improve the "binning" step: (1) k-mer frequencies that are somewhat preserved throughout a single microbial genome (*Pride et al., 2003*) to identify contigs that likely originate from the same genome (*Teeling et al., 2004*), and (2) a set of genes that occur in the vast majority of bacterial genomes as a single copy to estimate the level of completion and contamination of genome bins (*Wu & Eisen, 2008*; *Campbell et al., 2013*; *Parks et al., 2015*). These properties, along with differential coverage of contigs across multiple samples when such data exist, are routinely used to identify coherent microbial draft genomes in metagenomic assemblies (*Dick et al., 2009*; *Albertsen et al., 2013*; *Wu et al., 2014*; *Alneberg et al., 2014*; *Kang et al., 2015*; *Eren et al., 2015*).

On the other hand, researchers who study eukaryotic genomes generally focus on the recovery of a single organism, which, in most cases, simplifies the identification of the target genome in assembly results. However, sequences of bacterial origin can contaminate eukaryotic genome assemblies due to their occurrence in samples (*Chapman et al., 2010*; *Artamonova & Mushegian, 2013*), DNA extraction kits (*Salter et al., 2014*), or laboratory environments (*Laurence, Hatzis & Brash, 2014*; *Strong et al., 2014*). One of the major challenges of working with eukaryotic genomes is the extent of repeat regions that complicate the assembly process (*Richard, Kerrest & Dujon, 2008*). To optimize the assembly, researchers often employ multiple library preparations for sequencing (*Gnerre et al., 2010*; *Ekblom & Wolf, 2014*), which may increase the potential sources of post-DNA extraction contamination. Contaminants in assembly results can eventually contaminate public databases (*Merchant, Wood & Salzberg, 2014*), and impair scientific findings (*Artamonova et al., 2015*). The detection and removal of contaminants poses a major bioinformatics challenge. To identify undesired contigs in a genomic assembly, scientists can simply compare their assembly results to public sequence databases for positive hits to unexpected taxa (*Ekblom & Wolf, 2014*), use k-mer coverage plots to identify distinct genomes (*Percudani, 2013*), or employ scatter plots to partition contigs based on their GC-content and coverage (*Kumar et al., 2013*). However, advanced solutions developed for accurate identification of microbial genomes in complex metagenomic assemblies can leverage these approaches further, and offer enhanced curation options for eukaryotic assemblies.

The first release of a tardigrade genome by *Boothby et al. (2015)* demonstrates a striking example of the importance of careful screening for contaminants in eukaryotic genome assemblies. Tardigrades are microscopic animals occurring in a wide range of ecosystems and they exhibit extended capabilities to survive in harsh conditions that would be fatal to most animals (*Ramløv & Westh, 2001*; *Jönsson, Harms-Ringdahl & Torudd, 2005*;

*Jönsson et al., 2008*; *Horikawa et al., 2013*). Boothby and his colleagues generated a composite DNA sequencing dataset from a culture of the tardigrade *Hypsibius dujardini* by exploiting some of the best practices of high-throughput sequencing available today (*Boothby et al., 2015*). In their assembled tardigrade genome, the authors detected a large number of genes originating from bacteria, making up approximately one-sixth of the gene pool, and suggested that horizontal gene transfers (HGTs) could explain the unique ability of tardigrades to withstand extreme ranges of temperature, pressure, and radiation. However, *Koutsovoulos et al.*'s (*2016*) subsequent analysis of Boothby et al.'s assembly suggested that it contained extensive bacterial contamination, casting doubt on the extended HGT hypothesis. By applying two-dimensional scatterplots on their own raw assembly results, Koutsovoulos et al. also reported a curated draft genome of *H. dujardini*.

Here we re-analyzed the raw sequencing data generated by *Boothby et al. (2015)* and *Koutsovoulos et al. (2016)*, in combination with an independent RNA-Seq dataset generated by *Levin et al. (2016)* for *H. dujardini*. Using anvi'o, an analysis and visualization platform originally designed for the identification of bacterial genomes in metagenomic assemblies (*Eren et al., 2015*), we employed bacterial single-copy genes to assess the occurrence of bacterial genomes in the raw and curated assembly results, utilized k-mer frequencies and coverage values across multiple sequencing libraries to organize scaffolds, and visualized our findings in a single display.

## MATERIAL AND METHODS

### Genome assemblies, and raw sequencing data for DNA and RNA

*Boothby et al. (2015)* constructed three paired-end Illumina libraries (insert sizes of 0.3, 0.5 and 0.8 kbp) for 2 × 100 paired-end sequencing on a HiSeq2000, and six single-end long-read libraries (five Illumina Moleculo libraries sequenced by the Illumina "long read" DNA sequencing service, and one PacBio SMRT library sequenced using the P6-C4 chemistry and a 1 X 240 movie), which altogether provided a co-assembly of 252.5 Mbp. The tardigrade genome released by *Boothby et al. (2015)*, along with the nine sequencing data used for its assembly, are available at http://weatherby.genetics.utah.edu/seq_transf. Independently, *Koutsovoulos et al. (2016)* generated a 0.3 kbp insert library and a 1.1 kbp insert mate-pair library for 2 × 100 paired end sequencing on a HiSeq2000 that provided a co-assembly of 185.8 Mbp (nHd.1.0). These authors subsequently curated a 135 Mbp draft genome (nHd.2.3) by removing potential contamination and re-assembling filtered short reads (*Koutsovoulos et al., 2016*). The tardigrade raw assembly and curated draft genome released by *Koutsovoulos et al. (2016)* are available at http://badger.bio.ed.ac.uk/H_dujardini, and their two sequencing datasets are available from the ENA, under study accession PRJEB11910 .

### RNA-seq data

We obtained the RNA-seq data using the NCBI accession id PRJNA272543 (*Levin et al., 2016*). Briefly, Levin et al. isolated RNA from *H. dujardini* using the Trizol reagent (Invotrogen), constructed paired-end Illumina libraries according to the TruSeq RNA-seq protocol, and sequenced their cDNA libraries with a read length of 100 bp.

## Quality filtering and read mapping

We used illumina-utils (*Eren et al., 2013*) (available from http://github.com/meren/illumina-utils) for quality filtering of short Illumina reads using 'iu-filter-quality-minoche' script with default parameters, which implements the quality filtering described by *Minoche, Dohm & Himmelbauer (2011)*. Bowtie2 v2.2.4 (*Langmead & Salzberg, 2012*) with default parameters mapped all reads to the scaffolds, and we used samtools v1.2 (*Li et al., 2009*) to convert reported SAM files to BAM files.

## Overview of the anvi'o workflow

Our workflow with anvi'o to identify and remove contamination from a given collection of scaffolds consists of four main steps. The first step is the processing of the FASTA file of scaffolds to create an anvi'o contigs database (CDB). The resulting database holds basic information about each scaffold in the assembly (such as the k-mer frequency, or GC-content). The second step is the profiling of each BAM file with respect to the CDB we generated in the previous step. Each anvi'o profile describes essential statistics for each scaffold in a given BAM file, including their average coverage, and the portion of each scaffold covered by at least one read. The third step is the merging of all anvi'o profiles. The merging step combines all statistics from individual profiles, and uses them to compute hierarchical clusterings of scaffolds. The default organization of scaffolds is determined by the average coverage information from individual profiles, and the sequence composition information from the CDB. This organization makes it possible to identify scaffolds that distribute similarly across different library preparations. The final step is the visualization of the merged data on the anvi'o interactive interface. The anvi'o interactive interface provides a holistic perspective of the combined data, which allows the identification of draft genome bins, and removal of contaminants.

## Processing of scaffolds, and mapping results

We used anvi'o v1.2.2 (available from http://github.com/meren/anvio) to process scaffolds and mapping results, visualize the distribution of scaffolds, and identify draft genomes following the workflow outlined in the previous section, and detailed in *Eren et al. (2015)*. We created an anvi'o contigs database CDB for each scaffold collection using the 'anvi-gen-contigs-database' program with default parameters (where k equals 4 for k-mer frequency analysis). We then annotated scaffolds with myRAST (available from http://theseed.org/) and imported these results into the CDB using the program 'anvi-populate-genes-table' to store the information about the locations of open reading frames (ORFs) in scaffolds, and their taxonomical and functional inference. We profiled individual BAM files using the program 'anvi-profile' with a minimum contig length of 1 kbp, and the program 'anvi-merge' combined resulting profiles with default parameters. For the analysis of *Boothby et al. (2015)* assembly, we also profiled the RNA-Seq data published by *Levin et al. (2016)* to identify scaffolds with transcriptomic activity, and exported the table for proportion of each scaffold covered by transcripts using the script 'get-db-table-as-matrix.' We used the supplementary material published by *Boothby et al. (2015)* ("Dataset S1" in the original publication) to identify scaffolds with proposed HGTs. Finally, we used
the program 'anvi-interactive' to visualize the merged data, and identify genome bins. We included RNA-Seq results and scaffolds with HGTs into our visualization using the '--additional-layers' flag. To finalize the anvi'o generated SVG files for publication, we used Inkscape v0.91 (available from https://inkscape.org/).

### Predicting the number of bacterial genomes in an assembly

We used the occurrence of bacterial single-copy genes as a proxy to the expected number of bacterial genomes in a raw assembly or in a curated genome bin. First, we ran on each CDB generated in this study the anvi'o program 'anvi-populate-search-tables' to search using HMMer v3.1b2 (*Eddy, 2011*) for bacterial single-copy genes *Campbell et al. (2013)* published. Then, we used the anvi'o script 'gen-stats-for-single-copy-genes' to report the number of hits per single-copy gene as an array of integers from each CDB. We finally used mode (i.e., the most frequently occurring number) of this array as the expected number of complete bacterial genomes in a given collection of scaffolds. For additional discussion regarding the relevance of this metric to predict the number of bacterial genomes in an assembly, see the Supplemental Information 1. The script 'gen-stats-for-single-copy-genes' also used the R library 'ggplot' v1.0.0 (*R Development Core Team R, 2011*; *Ginestet, 2011*) to plot the occurrence of single-copy genes.

### Taxonomical and functional annotation of bacterial genomes

We uploaded bacterial draft genomes identified from the raw tardigrade genomic assembly results into the RAST server (*Aziz et al., 2008*), and used the RAST best taxonomic hits and FigFams to infer the taxonomy of genome bins and functions they harbor.

### Data availability

The URL http://merenlab.org/data/ reports (1) anvi'o files to regenerate Figs. 1 and 2, (2) our curation of the tardigrade genome from Boothby et al.'s assembly (which is also available through the NCBI under the bioproject ID PRJNA309530), and (3) the FASTA files for bacterial genomes we identified in the raw assemblies from Boothby et al. and Koutsovoulos et al.

## RESULTS AND DISCUSSION

*Boothby et al. (2015)* generated sequencing data from a tardigrade culture using three short read (Illumina) and six long read (Moleculo and PacBio) libraries, which altogether provided a co-assembly of 252.5 Mbp. Using this assembly, the authors suggested that 6,663 genes were entered into the tardigrade genome through HGTs. Independently, Koutsovoulos et al. generated sequencing data from another tardigrade culture using two short read Illumina libraries that provided a co-assembly of 185.8 Mbp, from which they could curate a 135 Mbp tardigrade draft genome by removing potential bacterial contamination using two-dimensional scatterplots of scaffolds with respect to their GC-content and coverage (*Koutsovoulos et al., 2016*).

### A holistic view of the data

The use of multiple library preparations and sequencing strategies is likely to result in more optimal assembly results (*Gnerre et al., 2010*). Hence, we focused on the scaffolds generated

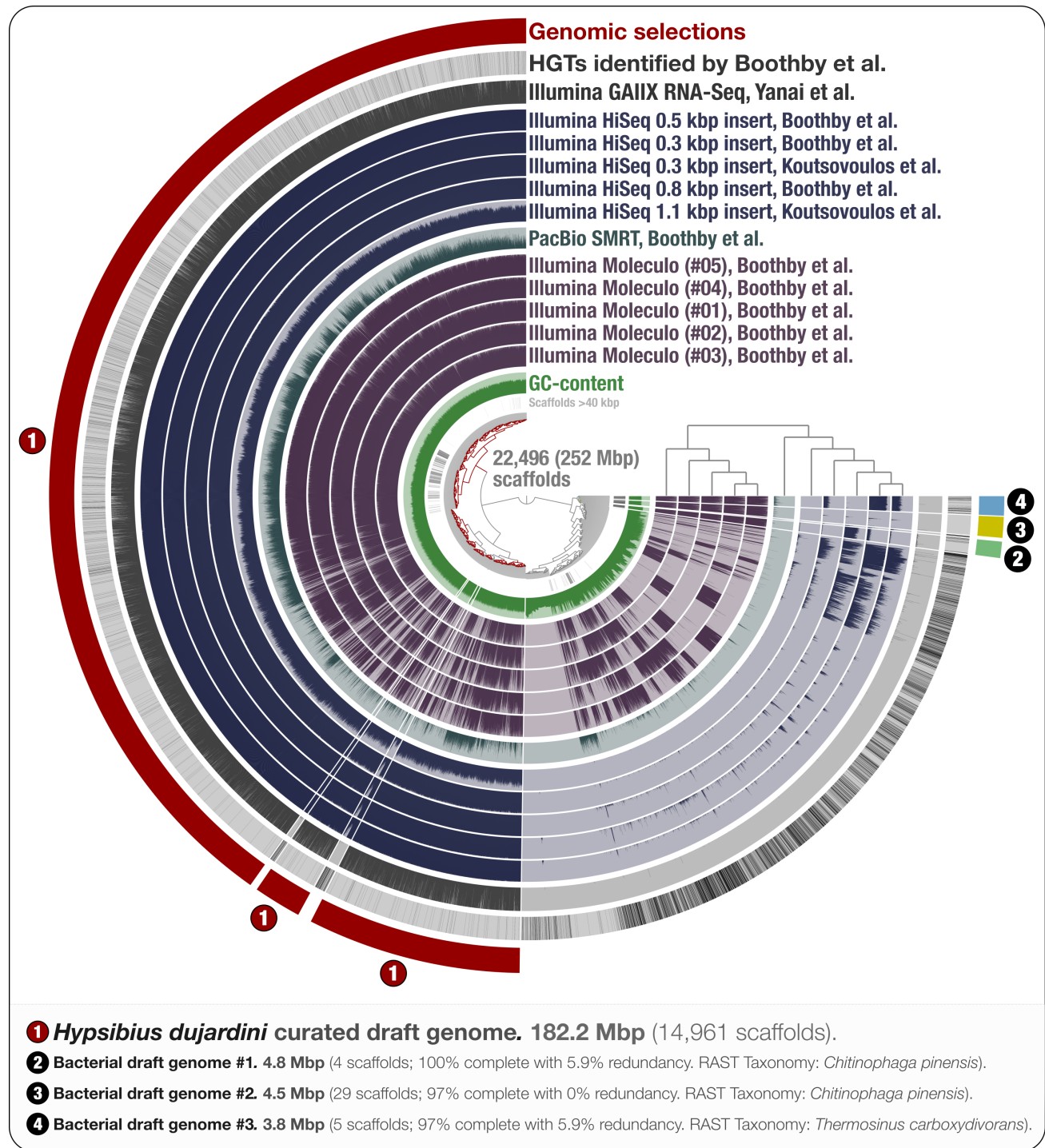

**Genomic selections**
**HGTs identified by Boothby et al.**
**Illumina GAIIX RNA-Seq, Yanai et al.**
**Illumina HiSeq 0.5 kbp insert, Boothby et al.**
**Illumina HiSeq 0.3 kbp insert, Boothby et al.**
**Illumina HiSeq 0.3 kbp insert, Koutsovoulos et al.**
**Illumina HiSeq 0.8 kbp insert, Boothby et al.**
**Illumina HiSeq 1.1 kbp insert, Koutsovoulos et al.**
**PacBio SMRT, Boothby et al.**
**Illumina Moleculo (#05), Boothby et al.**
**Illumina Moleculo (#04), Boothby et al.**
**Illumina Moleculo (#01), Boothby et al.**
**Illumina Moleculo (#02), Boothby et al.**
**Illumina Moleculo (#03), Boothby et al.**
**GC-content**
Scaffolds >40 kbp

22,496 (252 Mbp)
scaffolds

① *Hypsibius dujardini* **curated draft genome. 182.2 Mbp** (14,961 scaffolds).

② **Bacterial draft genome #1. 4.8 Mbp** (4 scaffolds; 100% complete with 5.9% redundancy. RAST Taxonomy: *Chitinophaga pinensis*).

③ **Bacterial draft genome #2. 4.5 Mbp** (29 scaffolds; 97% complete with 0% redundancy. RAST Taxonomy: *Chitinophaga pinensis*).

④ **Bacterial draft genome #3. 3.8 Mbp** (5 scaffolds; 97% complete with 5.9% redundancy. RAST Taxonomy: *Thermosinus carboxydivorans*).

**Figure 1** **Holistic assessment of the tardigrade genome assembly from *Boothby et al. (2015)*.** Dendrogram in the center organizes scaffolds based on sequence composition, and coverage values acquired from 11 DNA libraries. Scaffolds larger than 40 kbp were split into sections of 20 kbp for visualization purposes. Splits are displayed in the first inner circle and GC-content (0–71%) in the second circle. In the following 11 layers, each bar represents the portion of scaffolds covered by short reads in a given sample. The next layer shows the same information for RNA-Seq data. Scaffolds harboring genes used by Boothby et al. to support the expended HGT hypothesis is shown in the next layer. Finally, the outermost layer shows our selections of scaffolds as draft genome bins: the curated tardigrade genome (selection #1), as well as three near-complete bacterial genomes originating from various contamination sources (selections #2, #3, and #4).

by *Boothby et al. (2015)* as a foundation to maximize the recovery of the tardigrade genome. To provide a holistic understanding of the composite sequencing data generated by the two teams, we mapped the raw data from the nine DNA sequencing libraries from Boothby et al., and the two Illumina libraries from *Koutsovoulos et al. (2016)* on this assembly. Anvi'o generated a hierarchical clustering of scaffolds by combining the tetra-nucleotide frequency and coverage of each scaffold across the 11 DNA sequencing libraries (*Eren et al., 2015*). Besides visualizing the coverage of each scaffold in each sample, we highlighted scaffolds with HGTs identified by Boothby et al. on the resulting organization of scaffolds, and visualized RNA-seq mapping results. Figure 1 displays the anvi'o merged profile that represents all this information in a single display.

## A draft genome for *H. dujardini*

Through the anvi'o interactive interface we selected 14,961 scaffolds from the Boothby et al. assembly that recruited large number of short-reads in a consistent manner (Fig. 1). This 182.2 Mbp selection with consistent coverage (#1 in Fig. 1) represents our curation of the tardigrade draft genome from Boothby et al.'s assembly. The remaining 7,535 scaffolds, which total about 70 Mbp of the assembly, harbored 96.1% of HGTs identified by Boothby et al. These scaffolds recruited only 0.05% of the reads from the RNA-Seq data, highlighting the extent of contamination in the original assembly. This finding is in agreement with Koutsovoulos et al.'s findings; however, our curated draft genome from the Boothby et al.'s assembly is 47 Mbp larger than the draft genome released by *Koutsovoulos et al. (2016)*, most probably due to Boothby et al.'s inclusion of longer reads from Moleculo libraries. While the portion of scaffolds covered by RNA-Seq data suggests that this additional 47 Mbp still originate from the tardigrade genome, the biological relevance of this information (or lack thereof) for the characterization of the tardigrade genome falls outside of the scope of our study.

## The origin of bacterial contamination

Our mapping results indicate the presence of non-target sequences in the assembly that recruit reads only from long-read libraries. One interpretation could be that most of the contamination in Boothby et al.'s assembly originated from Moleculo libraries, post DNA-extraction (Fig. 1). However, while a recent study shows that the majority of long reads from Moleculo libraries originated from low-abundance organisms in the analyzed samples (*Sharon et al., 2015*), another study suggests relatively more sequencing bias in Moleculo library preparation results (*Kuleshov et al., 2015*). Therefore, an alternative interpretation of the mapping results can be that the bacterial contaminants were present in the sample pre-DNA extraction at very low abundances, and each Moleculo library preparation included long reads originating from different parts of this rare community. Regardless, long reads considerably improved Boothby et al.'s assembly, which resulted in a larger tardigrade genome following the removal of non-target sequences. While these results reiterate that the use of long-read libraries is essential to generate more comprehensive assemblies, they also suggest that extra care should be taken to better mitigate the presence of non-target sequences in assembly results when long-read libraries are used for sequencing.

We identified three near-complete bacterial genomes affiliated to *Chitinophaga* and *Thermosinus* in Boothby et al.'s assembly (Fig. 1). Surprisingly, Boothby et al. identified only a small portion of these complete bacterial genomes as sources of HGTs while applying a metric specifically designed to detect foreign DNA in eukaryotic genomes. For instance, none of the 4,459 genes in bacterial draft genome #2 (selection #3 in Fig. 1) were reported in Boothby et al.'s findings as HGTs. We also processed and visualized the raw assembly (nHd.1.0) from *Koutsovoulos et al. (2016)* using anvi'o (Fig. S1), and recovered eight bacterial genomes. However, we found no taxonomical overlap between high-completion bacterial genomes from the two sequencing projects (Table S1).

Interestingly, one bacterial genome (selection #2 in Fig. 1) was detected in DNA libraries from both groups, as well as in the RNA-seq data, suggesting that the related bacterial population was in all samples prior to the DNA/RNA extraction step. This genome is affiliated to *Chitinophaga*, and harbors genes coding for chitin degradation and utilization (Table S2). Chitin occurs naturally in the feeding apparatus of tardigrades (*Guidetti et al., 2015*), and might be a source of carbon for its microbial inhabitants. The genome also harbors genes coding for the biosynthesis of proteorhodopsin, host invasion and intracellular resistance, dormancy and sporulation, oxidative stress, and tryptophan, which is an essential amino acid for animals (*Crawford, 1989*; *Zelante et al., 2013*). Although this genome may belong to a tardigrade symbiont, the generation of the data does not allow us to rule out the possibility that it may be associated with the food source. Nevertheless, this finding suggests that there may be cases where non-target genomes in an assembly can provide clues about the lifestyle of a given host.

## Best practices to assess bacterial contamination

Initial assessment of the occurrence of bacterial single-copy genes can provide a quick estimation of the number of bacterial genomes that occur in assembly results (Supplemental Information 1). The use of bacterial single-copy genes can give much more accurate representation of potential bacterial contamination than screening for 16S rRNA genes alone, as they are less likely to be found in co-assembly results (*Miller et al., 2011*; *Delmont et al., 2015*). Although *Boothby et al. (2015)* reported the lack of 16S rRNA genes in their assembly, anvi'o estimated that it contained at least 10 complete bacterial genomes (Fig. 2) using a bacterial single-copy gene collection (*Campbell et al., 2013*). This simple yet powerful step could identify cases of extensive contamination, and alert researchers to be diligent in identifying scaffolds originating from bacterial organisms. Figure 2 also summarizes the HMM hits in scaffolds found in curated tardigrade genomes from our analysis and Koutsovoulos et al.'s study. We observed that the average significance score for the remaining HMM hits for bacterial single-copy genes in curated genomes was 4.2 times lower in average compared to the HMM hits in assembly results (Table S3). The decrease in the significance scores, and the very similar patterns of occurrence of HMM hits between the two curation efforts suggest that some of the HMM profiles may not be specific enough to be identified only in bacteria.

Two-dimensional scatterplots have a long history of identifying distinct genomes in assembly results (*Tyson et al., 2004*) and continue to be used for delineating microbial

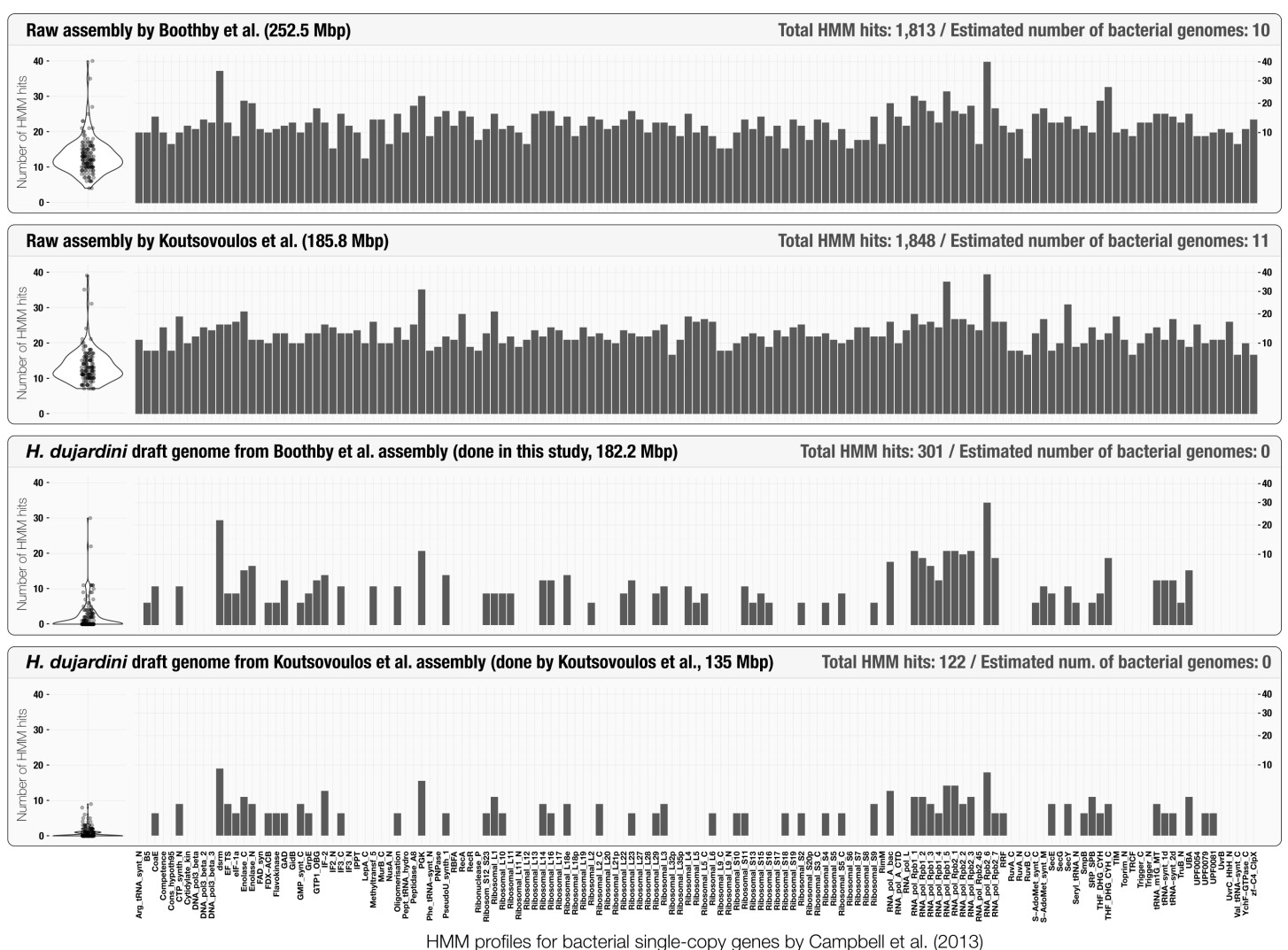

**Figure 2  Occurrence of the 139 bacterial single-copy genes reported by *Campbell et al. (2013)* across scaffold collections.**  The top two plots display the frequency and distribution of single-copy genes in the raw tardigrade genomic assembly generated by *Boothby et al. (2015)*, and *Koutsovoulos et al. (2016)*, respectively. The bottom two plots display the same information for each of the curated tardigrade genomes. Each bar represents the squared-root normalized number of significant hits per single-copy gene. The same information is visualized as box-plots on the left side of each plot.

genomes in metagenomic assemblies (*Albertsen et al., 2013*; *Cantor et al., 2015*), as well as detecting contamination in eukaryotic assembly results (*Kumar et al., 2013*). Although scatterplots can describe the organization of assembled contigs, they suffer from limited number of dimensions they can display, and their inability to depict complex supporting data that can improve the identification of individual genomes. These limitations are particularly problematic in sequencing projects covering multiple sequencing libraries, where displaying mapping results from each library can help detecting sources of contaminants. Despite their successful applications, two dimensional scatter plots limit researchers to the use of simple characteristics of the data that can be represented on an axis (such as GC-content). In contrast, clustering scaffolds, and overlaying multiple

layers of independent information produce more comprehensive visualizations that display multiple aspects of the data.

## CONCLUSIONS

The field of genomics requires advanced computational approaches to take best advantage of constantly evolving ways to generate sequencing data, and to identify and remove contamination from genome assemblies. Our study indicates that some of these advanced approaches may emerge from the field of metagenomics, where the need for *de novo* reconstruction of microbial genomes from environmental samples has given raise to techniques and software platforms that can make sense of complex assemblies. Here we used k-mer frequencies to organize scaffolds, the occurrence of bacterial single-copy genes to estimate the extent of contamination, and an advanced visualization strategy to detect and remove contamination in a eukaryotic assembly project while simultaneously characterizing the sources of contamination. Our results also suggest that metagenomic binning strategies can be used to recover near-complete bacterial genomes from raw eukaryotic assemblies, which can provide insights into the potential host-microbe interactions during the curation step.

## ACKNOWLEDGEMENTS

We are grateful to Thomas C. Boothby, Georgios Koutsovoulos, Sujai Kumar, and their colleagues for making their data available and answering our questions. We thank Itai Yanai for providing us with the RNA-Seq data ahead of publication. We also thank Hilary G. Morrison for her invaluable suggestions. We finally thank our editor and reviewers for their valuable comments and suggestions.

### Funding

This work was supported by the Frank R. Lillie Research Innovation Award, and startup funds from the University of Chicago. The funders had no role in study design, data collection and analysis, decision to publish, or preparation of the manuscript.

### Grant Disclosures

The following grant information was disclosed by the authors:
Frank R. Lillie Research Innovation Award.
University of Chicago.

### Competing Interests

A. Murat Eren is an Academic Editor for PeerJ.

### Author Contributions

- Tom O. Delmont and A. Murat Eren conceived and designed the experiments, performed the experiments, analyzed the data, contributed reagents/materials/analysis tools, wrote the paper, prepared figures and/or tables, reviewed drafts of the paper.

## Data Availability

http://merenlab.org/data/.

## Supplemental Information

Supplemental information for this article can be found online at http://dx.doi.org/10.7717/peerj.1839#supplemental-information.

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
