# Peer review of "Identifying contamination with advanced visualization and analysis practices: metagenomic approaches for eukaryotic genome assemblies"

_PeerJ, doi:10.7717/peerj.1839_

## Round 0.1 · original submission · Major Revisions

· Academic Editor

Major Revisions

The reviewers are generally supportive, but also make a few pertinent points that need to be addressed before publication. In particular, both mention the need for more details on anvi'o. Given that the paper does not introduce a that method, I however don't think that a comparison with another pipeline is necessary.

Also, please provide more methodological details on the RNA-seq data (sample prep, etc).

·

Basic reporting

The authors make use of a previously published software pipeline for metagenomic analysis together with RNA-seq data to quality check and improve genome assembly, using as their test case the tardigrade, concluding that contamination, particularly deriving from use of some technologies, underlie most of its previously suggested high prevalence of bacterial-to-eukaryote HGT. Basic approach of MG community analysis method to validate purity of genome sequencing assembly is elegant and needs to be published, and the result that RNA-seq invalidates many of the putative HGTs in the first published tardigrade genome is also valuable, as is the finding of a putative symbiont and the highlighting that a particular library preparation method may be prone to amplifying contaminants that current foreign DNA filters have difficulties identifying.

See change requests below in comments to author section.

Experimental design

This is fine, but the use of the software must be made clearer so that this can be easily assessed by the reader.

Validity of the findings

This is fine.

Additional comments

Requests:
- I don't see anywhere where the RNA-seq data has been deposited, nor where the new corrected genome has been deposited.
- Abstract and elsewhere claims anvi'o is routinely used. Please substantiate this - I had never heard of it before.
- lines 45-50 outline one type of metagenomic workflow but misses the alternative approach used e.g. in large-scale human gut studies; building gene catalogs/genome collections, then mapping reads directly to them. This also should be at least mentioned as a binning-free approach, even though it is not directly relevant to the work at hand.
- Methods generally: even if anvi'o has been published, few probably know the details of how it functions. For this reason methods need to be clearer on what exactly ends up being done by choosing particular options in the software, in order to understand the nature of the results. For the paper to be self-contained, there needs to be more background on what the software does.
-- For example, it is not clear how the estimation of number of genomes is done. If there are 139 genes semi-universally single-copy, and HMMER is used to find their matches in samples, how does that yield number of distinct genomes? It would e.g. if the hits were then clustered (or the queries first clustered somehow), so as to check how many clusters there are per gene family, but the section does not state. This needs to be made clear.
-- For classifications of genome bins likewise: are all reads in each bin uploaded to RAST and assessed, or is there some redundancy removal first?

·

Basic reporting

The paper appears to me well written, complete and easy understandable.

Experimental design

The detection of contamination in reference genomes or more broadly in de novo assemblies is still a huge problem. There have been methods suggested like Kraken to perform such a search, but it appears to me that the field is still very young.
The here presented work aims at two things. First, it reports an improved de novo assembly for Hypsibius dujardini by combining previous published data. Second, and maybe more importantly, it describes a possibility on how to detect contamination. While the methods are well described I personally have a bit of a problem to identify the main massage of the paper.
The abstract and also the introduction reads like that the authors present an improved genome assembly of Hypsibius dujardini. Later on, the authors focuses on the methods to detect and assess the contamination. The conclusion, however, focuses on de novo assembling of bacterial genomes out of metagenomics data. Thus, I would recommend to make it clearer. I think the main contribution to the field might be the summary of the methods the authors used to detect the contamination. Furthermore, it would be nice to see how the methods used in this paper perform in comparison to other. Thus, I would suggest to run the obtained and previous assemblies through methods like Kraken or other methagenomic analysis methods. This was actually suggested in an article (Merchant et al., 2014) to detect bacterial contamination.

Validity of the findings

The findings and the way they are obtained are robust and controlled.

Additional comments

I enjoyed reading your paper. I would encourage you to focus more on the methods that you described and compare them to existing approaches to detect contamination.

---

## Round 0.2 · Minor Revisions

· Academic Editor

Minor Revisions

Thank you for your resubmission. I had a look at your point-by-point responses and changes in the manuscript. I am satisfied with them. In particular, I thought that the additional plots you provided in support of your approach to count the number of genomes were helpful. However, I feel that you should add them to the paper, as readers may have the same question as Reviewer #1.

Could you please do this small change? This will clear the way for publication.

---

## Author Rebuttal · Round 0.2

A. Murat Eren, Ph.D.

ASSISTANT PROFESSOR, DEPARTMENT OF MEDICINE

**Knapp Center for Biomedical Discovery**
900 E. 57th Street, Mailbox 9, Room 9118, Chicago, IL 60637
P: +1-773-702-5935 / F: +1-773-702-2281 / meren@uchicago.edu

# Response to the Reviewers

We thank the editor and reviewers for their time and interest in our study. We now have made the necessary changes in the manuscript to address each point raised by the editor, and the reviewers.

Thanks to the availability of our pre-print, we also had the opportunity to collect suggestions from the community. We included further changes in our revised manuscript to address some of the suggestions we received. These changes include being more specific about the raw and curated assemblies by Koutsovoulos et al. by using the same versioning strategy employed by these authors (nHd.1.0 and nHd.2.3), and avoiding any qualitative suggestions regarding the different versions of tardigrade genomes.

Our responses to the editor, and our reviewers follow.

## *The Editor*

***The reviewers are generally supportive, but also make a few pertinent points that need to be addressed before publication. In particular, both mention the need for more details on anvi'o. Given that the paper does not introduce a method, I however don't think that a comparison with another pipeline is necessary. Also, please provide more methodological details on the RNA-seq data (sample prep, etc).***

We have expanded the Materials and Methods section to include a description of the anvi'o workflow, and the details of the RNA-Seq data.

## *Reviewer #1*

**BASIC REPORTING**

***The authors make use of a previously published software pipeline for metagenomic analysis together with RNA-seq data to quality check and improve genome assembly, using as their test case the tardigrade, concluding that contamination, particularly deriving from use of some technologies, underlie most of its previously suggested high prevalence of bacterial-to-eukaryote HGT. Basic approach of MG community analysis method to***

[Figure]

A. Murat Eren, Ph.D.
ASSISTANT PROFESSOR, DEPARTMENT OF MEDICINE

**Knapp Center for Biomedical Discovery**
900 E. 57th Street, Mailbox 9, Room 9118, Chicago, IL 60637
P: +1-773-702-5935 / F: +1-773-702-2281 / meren@uchicago.edu

*validate purity of genome sequencing assembly is elegant and needs to be published, and the result that RNA-seq invalidates many of the putative HGTs in the first published tardigrade genome is also valuable, as is the finding of a putative symbiont and the highlighting that a particular library preparation method may be prone to amplifying contaminants that current foreign DNA filters have difficulties identifying.*

### EXPERIMENTAL DESIGN

*This is fine, but the use of the software must be made clearer so that this can be easily assessed by the reader.*

> We thank the reviewer for their suggestion. The manuscript now describes the anvi'o workflow in greater details under the Material and Methods section.

### COMMENTS FOR THE AUTHOR

*I don't see anywhere where the RNA-seq data has been deposited, nor where the new corrected genome has been deposited.*

> The genome we curated is submitted to NCBI under bioproject number PRJNA309530 (http://www.ncbi.nlm.nih.gov/bioproject/309530), but the NCBI staff members are still processing the genome according to their last e-mail to us on Feb 22nd, 2016. Anticipating the long processing time, we already made this genome (as well as the bacterial genomes we identified) to the community via http://merenlab.org/data/.

> RNA-Seq dataset is available from NCBI under the study accession PRJNA272543. We have now added more information in the text to explain the details of RNA-Seq data generation.

*Abstract and elsewhere claims anvi'o is routinely used. Please substantiate this - I had never heard of it before.*

> We apologize for the lack of clarity there. The "routinely used approaches" statement in the abstract in fact referred to are approaches that are common to the assembly-based metagenomic workflow (such as the use of bacterial single-copy genes, organization of contigs based on sequence composition), and we showed the relevance of these approaches throughout the manuscript with references. However, we understand the lack of clarity, and decided to take that statement out of the abstract to avoid potential misunderstandings.

*lines 45-50 outline one type of metagenomic workflow but misses the alternative approach used e.g. in large-scale human gut studies; building gene catalogs/genome collections,*

[Figure]

A. Murat Eren, Ph.D.
ASSISTANT PROFESSOR, DEPARTMENT OF MEDICINE

**Knapp Center for Biomedical Discovery**
900 E. 57th Street, Mailbox 9, Room 9118, Chicago, IL 60637
P: +1-773-702-5935 / F: +1-773-702-2281 / meren@uchicago.edu

*then mapping reads directly to them. This also should be at least mentioned as a binning-free approach, even though it is not directly relevant to the work at hand.*

We acknowledge importance of generating gene catalogs to metagenomic studies in general when genome binning is not feasible. This approach has been particularly useful for large metagenomic projects such as METAHIT, and TARA Oceans. However, as the reviewer also pointed out, gene catalogs are not directly relevant to the curation of eukaryotic genome assemblies. To keep the introduction focused, we did not include many approaches that are very useful for metagenomic studies while they are not as useful for eukaryotic assembly projects. We respectfully decided to continue to reflect this in our introduction.

*Methods generally: even if anvi'o has been published, few probably know the details of how it functions. For this reason methods need to be clearer on what exactly ends up being done by choosing particular options in the software, in order to understand the nature of the results. For the paper to be self-contained, there needs to be more background on what the software does.*

We thank the reviewer for pushing us to do this! The manuscript now includes a more detailed description of the anvi'o workflow under the Materials and Methods section.

*For example, it is not clear how the estimation of number of genomes is done. If there are 139 genes semi-universally single-copy, and HMMER is used to find their matches in samples, how does that yield number of distinct genomes? It would e.g. if the hits were then clustered (or the queries first clustered somehow), so as to check how many clusters there are per gene family, but the section does not state. This needs to be made clear.*

We apologize for the lack of clarity with this. We re-wrote the text regarding this in the methods section to improve clarity. Briefly, once we have identified the occurrence of each single-copy gene in a collection of scaffolds using previously-published HMM profiles, we use the most frequently occurring number among hits as the expected number of bacterial genomes to be found in this collection. In other words, this is the 'mode' of the array of the number of single-copy gene hits. We would like to provide more information to the editor and our reviewers about the relevance of this metric to predict the number of genomes in an assembly by showing examples from other datasets.

## THE UNIVERSITY OF CHICAGO MEDICINE

A. Murat Eren, Ph.D.
ASSISTANT PROFESSOR, DEPARTMENT OF MEDICINE

**Knapp Center for Biomedical Discovery**
900 E. 57th Street, Mailbox 9, Room 9118, Chicago, IL 60637
P: +1-773-702-5935 / F: +1-773-702-2281 / meren@uchicago.edu

[Figure]

The figure shows the distribution of the number of hits for each bacterial single-copy gene collection published by three groups (including the one from Campbell et al., which we used in our study) for a raw metagenomic assembly of human gut microbiome. The mode of each array of hits predicts 10, 9, and 10 bacterial genomes respectively. A detailed analysis of this dataset by Sharon et al. (doi: 10.1101/gr.142315.112), and our group (doi: 10.7717/peerj.1319) identified 12 bacterial draft genomes, 8 of which were complete or near complete.

As a control, the same approach correctly predicts 1 genome in the raw assembly results of a cultivar:

[Figure]

Finally, predictions from these three single-copy gene collections show remarkable stability in very complex datasets:

A. Murat Eren, Ph.D.
ASSISTANT PROFESSOR, DEPARTMENT OF MEDICINE

**Knapp Center for Biomedical Discovery**
900 E. 57th Street, Mailbox 9, Room 9118, Chicago, IL 60637
P: +1-773-702-5935 / F: +1-773-702-2281 / meren@uchicago.edu

[Figure]

The particular analysis shown in this figure was done an the raw assembly results generated from an ocean sample, where the mode of three different collections predict the occurrence of 451, 451, and 431 genomes in this assembly.

Although it is quite a simple metric, our results suggest that it offers a somewhat reliable first approximation to the expected number of bacterial genomes in an assembly.

***For classifications of genome bins likewise: are all reads in each bin uploaded to RAST and assessed, or is there some redundancy removal first?***

RAST (please note that it is not MG-RAST) is a platform to characterize assembled genomes for functional and taxonomical characterization (http://rast.nmpdr.org/). Hence, we did not upload short reads, but only scaffolds in our draft genome bins. We have now clarified this in the methods section to avoid any confusion.

We are thankful to Dr. Forslund for their interest in our study, and for their valuable comments.

## THE UNIVERSITY OF CHICAGO MEDICINE

A. Murat Eren, Ph.D.
ASSISTANT PROFESSOR, DEPARTMENT OF MEDICINE

**Knapp Center for Biomedical Discovery**
900 E. 57th Street, Mailbox 9, Room 9118, Chicago, IL 60637
P: +1-773-702-5935 / F: +1-773-702-2281 / meren@uchicago.edu

## *Reviewer #2*

**BASIC REPORTING**

*The paper appears to me well written, complete and easy understandable.*

**EXPERIMENTAL DESIGN**

*The detection of contamination in reference genomes or more broadly in de novo assemblies is still a huge problem. There have been methods suggested like Kraken to perform such a search, but it appears to me that the field is still very young.*

*The here presented work aims at two things. First, it reports an improved de novo assembly for Hypsibius dujardini by combining previous published data. Second, and maybe more importantly, it describes a possibility on how to detect contamination. While the methods are well described I personally have a bit of a problem to identify the main massage of the paper. The abstract and also the introduction reads like that the authors present an improved genome assembly of Hypsibius dujardini. Later on, the authors focuses on the methods to detect and assess the contamination. The conclusion, however, focuses on de novo assembling of bacterial genomes out of metagenomics data. Thus, I would recommend to make it clearer. I think the main contribution to the field might be the summary of the methods the authors used to detect the contamination.*

> We thank the reviewer for pointing this out. Thanks to this comment, we realized that our concluding statements are not reflecting our focus, and our contribution. We now have changed it to make sure it reflects our original intention, how approaches utilized for the assembly-based metagenomics workflow for environmental samples can be used to enhance the detection and removal of contaminants in eukaryotic genome assemblies.

*Furthermore, it would be nice to see how the methods used in this paper perform in comparison to other. Thus, I would suggest to run the obtained and previous assemblies through methods like Kraken or other methagenomic analysis methods. This was actually suggested in an article (Merchant et al., 2014) to detect bacterial contamination.*

> We also thank the reviewer for referring to Kraken. We are aware of this tool and its capabilities (which similar to Phymmbl, CLARK, myRAST, and other tools that perform taxonomical annotation of metagenomic short reads and/or contigs). We in fact refer to the taxonomy-based approach (and also to the Merchant et al. paper) in our introduction. However, Kraken only provides taxonomical information, and it cannot differentiate

[Figure]

A. Murat Eren, Ph.D.

ASSISTANT PROFESSOR, DEPARTMENT OF MEDICINE

**Knapp Center for Biomedical Discovery**
900 E. 57th Street, Mailbox 9, Room 9118, Chicago, IL  60637
P: +1-773-702-5935 / F: +1-773-702-2281 / meren@uchicago.edu

between true HGTs and true contaminants. Furthermore, a screening approach based on taxonomical annotations will leave behind bacterial contigs that did not get annotated in the assembly. In our study we used myRAST for taxonomical inference of contigs. However, we used this information only as subsidiary information, and instead, we relied upon coverage values across samples, coupled with k-mer frequencies, RNA-Seq data, and the occurrence of single copy genes to identify contamination. Using this multifaceted approach on the tardigrade genome, we could reconstruct entire bacterial genomes, which include contigs with no taxonomical hits. Taxonomical inferences alone could not have detected all contigs of bacterial origin we identified due to a lack of reference genomic databases. Kraken is a useful tool to assign taxonomy, it is not designed to identify and/or remove contaminants due to apparent limitations of sensitivity of taxonomical annotation for this task.

## VALIDITY OF THE FINDINGS

***The findings and the way they are obtained are robust and controlled.***

## COMMENTS FOR THE AUTHOR

***I enjoyed reading your paper. I would encourage you to focus more on the methods that you described and compare them to existing approaches to detect contamination.***

We are thankful for the helpful comments, and the interest. We hope that by changing our conclusion for a more focused summary of our study besides other improvements in the manuscript will bring forward the appropriateness of using methods developed for the assembly-based metagenomic workflow for characterizing contamination in eukaryotic genome assemblies.

---

## Round 0.3 · accepted · Accept

· Academic Editor

Accept

Thank you for the change. This clears the way for publication.

---

## Author Rebuttal · Round 0.3

THE UNIVERSITY OF
CHICAGO
MEDICINE

A. Murat Eren, Ph.D.
ASSISTANT PROFESSOR, DEPARTMENT OF MEDICINE

**Knapp Center for Biomedical Discovery**
900 E. 57th Street, Mailbox 9, Room 9118, Chicago, IL 60637
P: +1-773-702-5935 / F: +1-773-702-2281 / meren@uchicago.edu

# Response to the Editor

*Thank you for your resubmission. I had a look at your point-by-point responses and changes in the manuscript. I am satisfied with them. In particular, I thought that the additional plots you provided in support of your approach to count the number of genomes were helpful. However, I feel that you should add them to the paper, as readers may have the same question as Reviewer #1.*

We have now extended our submission with a three-page supplementary information file with figures we used to elucidate the relevance of our approach to predict the number of bacterial genomes in an assembly. We also added a sentence in the Materials and Methods section to refer the reader to this information.

We believe this was a very meaningful addition to our study. Not only it helped us to write a short but coherent text to introduce the idea to the community in a more formal way, but it will also help us to point this as a resource in the future. We thank the editor for their time and valuable suggestion that clearly improved this manuscript.